# BOSO: A novel feature selection algorithm for linear regression with high-dimensional data

**Luis V. Valcárcel**[1,2], **Edurne San José-Enériz**[2,3], **Xabier Cendoya**[1], **Ángel Rubio**[1,4,5], **Xabier Agirre**[2,3], **Felipe Prósper**[2,3,6,7], **Francisco J. Planes**[1,4,5]*

**1** Universidad de Navarra, Tecnun Escuela de Ingeniería, San Sebastián, Spain, **2** Universidad de Navarra, CIMA Centro de Investigación de Medicina Aplicada, Pamplona, Spain, **3** CIBERONC Centro de Investigación Biomédica en Red de Cáncer, Pamplona, Spain, **4** Universidad de Navarra, Centro de Ingeniería Biomédica, Pamplona, Spain, **5** Universidad de Navarra, DATAI Instituto de Ciencia de los Datos e Inteligencia Artificial, Pamplona, Spain, **6** IdiSNA Instituto de Investigación Sanitaria de Navarra, Pamplona, Spain, **7** Clínica Universidad de Navarra, Pamplona, Spain

* fplanes@tecnun.es

**Data Availability Statement:** All relevant data supporting findings of the study are within the manuscript and its Supporting Information files. The code used to generate the results shown in

## Abstract

With the frenetic growth of high-dimensional datasets in different biomedical domains, there is an urgent need to develop predictive methods able to deal with this complexity. Feature selection is a relevant strategy in machine learning to address this challenge. We introduce a novel feature selection algorithm for linear regression called BOSO (Bilevel Optimization Selector Operator). We conducted a benchmark of BOSO with key algorithms in the literature, finding a superior accuracy for feature selection in high-dimensional datasets. Proof-of-concept of BOSO for predicting drug sensitivity in cancer is presented. A detailed analysis is carried out for methotrexate, a well-studied drug targeting cancer metabolism.

## Author summary

We present BOSO (Bilevel Optimization Selector Operator), a novel method to conduct feature selection in linear regression models. In machine learning, feature selection consists of identifying the subset of input variables (features) that are correctly associated with the response variable that is aimed to be predicted. An adequate feature selection is particularly relevant for high-dimensional datasets, commonly encountered in biomedical research questions that rely on -omics data, *e.g.* predictive models of drug sensitivity, resistance or toxicity, construction of gene regulatory networks, biomarker selection or association studies. The need of feature selection is emphasized in many of these complex problems, since the number of features is greater than the number of samples, which makes it harder to obtain accurate and general predictive models. In this context, we show that the models derived by BOSO make a better combination of accuracy and simplicity than competing approaches in the literature. The relevance of BOSO is illustrated in the prediction of drug sensitivity of cancer cell lines, using RNA-seq data and drug screenings from GDSC (Genomics of Drug Sensitivity in Cancer) database. BOSO obtains linear regression models with a similar level of accuracy but involving a substantially lower number of features, which simplifies the interpretation and validation of predictive models.

this article can be found on GitHub (https://github.com/lvalcarcel/BOSO). It is distributed under the GNU General Public License (version 3).

**Funding:** This work was supported by the Minister of Economy and Competitiveness of Spain [PID2019-110344RB-I00, F.J.P.], PIBA Programme of the Basque Government [PIBA_2020_01_0055, F.J.P.], Instituto de Salud Carlos III (ISCIII) [PI16/02024, PI17/00701, F.P.], CIBERONC (Co-financed with European Union FEDER funds) [CB16/12/00489, F.P.], ERANET program ERAPerMed [MEET-AML, F.P.], MINECO Explora [RTHALMY, F.P.], Elkartek programme of the Basque Government [KK-2020/00008, F.J.P.], Cancer Research UK and AECC under the Accelerator Award Programme [C355/A26819, F.P.] and Fundación Ramon Areces [PREMMAN, F.J.P.]. L.V.V. received his salary from PFIS [FI17/00297] award from Instituto de Salud Carlos III (ISCIII). X.C. received his salary from Basque Government predoctoral grant [PRE_2018.2.0297]. The funders had no role in study design, data collection and analysis, decision to publish, or preparation of the manuscript.

**Competing interests:** The authors have declared that no competing interests exist.

## Introduction

High-dimensional datasets are currently an essential part of biomedical research [1–3]. Much effort has been devoted to developing statistical and machine learning methods able to deal with this complexity and avoid overfitting in problems with a limited sample size [4–8]. Dimensionality reduction and feature selection are the most commonly used strategies to address this issue [9,10]. Feature selection, which consists of identifying the true explanatory variables over the entire set of variables, have been extensively applied to both supervised and unsupervised learning problems [11].

Different feature selection approaches can be found for linear regression models, aimed to explain a response (dependent) variable as a linear combination of a set of input (independent) variables. The most popular feature selection algorithm is the Lasso regression [12], which is implemented in different machine learning software packages and integrated in dozens of algorithms for a varied range of biological questions [13–16]. However, as recently shown in Hastie et al. 2017 [17], the Lasso regression still has substantial room for improvement in high-dimensional datasets. In that work, using synthetic data in a number of conditions, the capacity of several approaches to elucidate the subset of variables that were used to generate the response variable was compared. In particular, they compared Lasso with a recent formulation of the best subset selection approach [18], which directly addresses the combinatorial problem of identifying the subset of features that more accurately fits the response variable through linear regression. They found that neither approach was significantly better than the other. Interestingly, they concluded that Relaxed Lasso [19], which combines the solution of Lasso and ordinary linear regression, incorporates the best of both approaches and is, therefore, the most accurate strategy in the literature.

Here, we propose a novel feature selection approach for linear regression called BOSO (Bilevel Optimization Selector Operator). We show that our approach is more accurate than Relaxed Lasso in many cases, particularly in high-dimensional datasets. Proof-of-concept of our approach is applied to predict drug sensitivity in cancer based on RNA-seq data. In particular, a detailed computational and *in-vitro* experimental analysis is presented for methotrexate, a well-studied drug targeting cancer metabolism [20].

## Results

### The BOSO algorithm

In linear regression, the best subset selection problem addresses the identification of variables correctly related with the response variable. This problem is presented here as a bilevel optimization problem and, for this reason, we call our approach Bilevel Optimization Selector Operator (BOSO). In particular, starting from a total set of $p$ features, BOSO searches for the best combination of features of length $K$ by solving a bilevel optimization problem, where the outer layer minimizes the validation error and the inner layer uses training data to minimize the loss function of the linear regression approach considered. Here, we chose Ridge regression for the training problem in order to account for multicollinearity in a simpler manner than Lasso; however, the formulation is also presented for ordinary linear regression (see Methods section for details).

In particular, BOSO relies on the observation that the optimal solution of the inner problem can be written as a set of linear equations that depends on the selected features. This observation makes it possible to solve a complex bilevel optimization problem via Mixed-Integer Quadratic Programming (MIQP) (see Methods section). This process is repeated for different $K$ values until an information criterion is not further improved. Here, we considered the Akaike

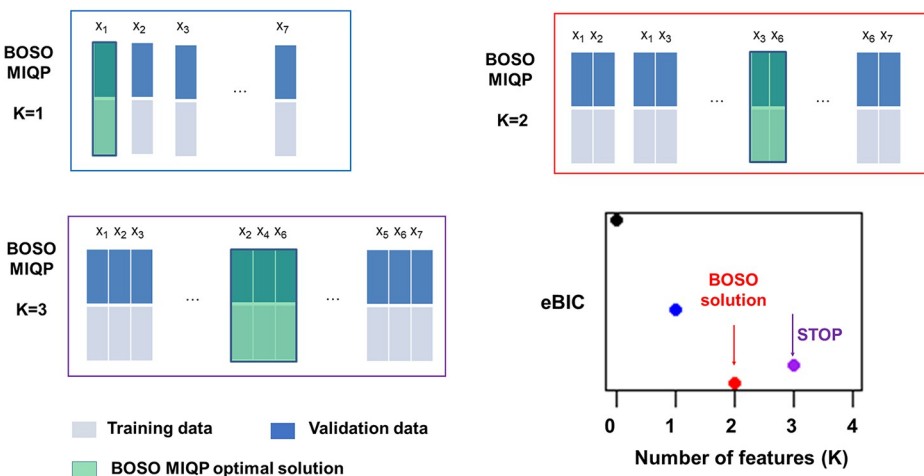

**Fig 1. Summary of the BOSO algorithm.** An example dataset with 7 features is split into training and validation sets. For any given subset of features of length *K*, a linear model is constructed with training data and assessed with validation data. The optimal selected features for a specific *K* value (green boxes) are obtained from the model that minimizes the validation error. For example, for *K* = 2, the linear model trained with the subset of features {$X_3$, $X_6$} is the one that minimizes the validation error. The problem of selecting the best subset of features of length *K* is formulated via mixed-integer quadratic programming (MIQP) (see Methods section) and solved using standard MIQP tools. With our MIQP approach, we directly assess all different combinations of linear models that involve *K* features and select the one with least validation error. This process is repeated for each *K* value until an information criterion, in this case the extended Bayesian Information Criterion (eBIC), is not further improved. Minimal eBIC is found in this example for *K* = 2. The final model is derived from Ridge regression with only these two selected variables.

Information Criterion (AIC) [21], the Bayesian Information Criterion (BIC) [22] and the extended BIC (eBIC) [23], which generalizes BIC when $p > n$, a common scenario in biomedical applications [24]. These were adjusted to take into account the use of Ridge regression instead of ordinary linear regression (see Methods section). Note here that other approaches use validation data to select the optimal *K*; instead, BOSO uses validation data to select the best subset of features of length *K*, and it uses the information criterion to select the optimal *K*. A conceptual scheme of BOSO for 7 variables can be found in Fig 1.

The core MIQP of BOSO addresses a hard-combinatorial optimization problem, whose complexity exponentially grows as *p* increases. Current MIQP solvers have been widely developed in the last decade [25]; however, in the case of BOSO, for large problems, they could take long computation times to guarantee optimality. This is also the case for the MIQP approach presented in Bertsimas et al, 2016 [18], referred to here as Best Subset. Here, we alleviated this issue by iteratively applying BOSO to random blocks of features of length *L* until convergence (see Methods section and S1 Fig). With this strategy, we substantially reduced the computation time of our approach and managed to apply BOSO to complex problems.

## Benchmarking of feature selection approaches

In order to assess the performance of BOSO, we replicated the same analysis presented in Hastie et al. 2017 [17], where relevant feature selection strategies, including Best Subset [18], Forward Stepwise [26,27], Lasso [12] and Relaxed Lasso [19], were compared. In that work, they generated synthetic data from a multivariate normal distribution in different settings, which depends on the number of instances, *n*; number of total available features, *p*; actual number of features contributing to the outcome, defined by the sparsity level *s* and their value (beta-type); covariance matrix between features $\Sigma_{ij} = \rho^{|i-j|}$, where $\rho$ is the autocorrelation level; and signal-

to-noise ratio (SNR level) (see S1 Appendix for further details). In particular, they considered 4 problem settings: low ($n = 100$, $p = 10$, $s = 5$), medium ($n = 500$, $p = 100$, $s = 5$), high-5 ($n = 50$, $p = 1000$, $s = 5$) and high-10 ($n = 100$, $p = 1000$, $s = 10$). These four problem settings were analyzed for different beta-types, autocorrelation level and signal-to-noise ratio.

In particular, we present here the results for one of the scenarios considered: beta-type 1, where the $s$ contributing features occur at (approximately) equally-spaced indices between 1 and $p$ with value 1, the remaining features being equal to 0; and an autocorrelation level between features of 0.35. In this beta-type, actual features contributing to the outcome show little correlation between each other. We tested the same levels of SNR analyzed in Hastie et al. 2017 [17], namely ten values of SNR from 0.05 to 6.00, equally distributed in logarithmic scale. In order to compare the capacity of different methods to extract the actual features contributing to the outcome, we used the F1-score, which is the harmonic mean of the precision and recall, the number of estimated non-zeros coefficients and the number of false positives and false negatives, metrics previously used in Hastie et al, 2017 [17] (see Methods section). We also included details as to other cases and evaluation metrics in S2–S21 Figs.

F1-scores obtained with BOSO, Lasso, Relaxed Lasso, Best Subset and Forward Stepwise in different cases are shown in Fig 2. For the Low setting ($p = 10$), BOSO performed slightly better

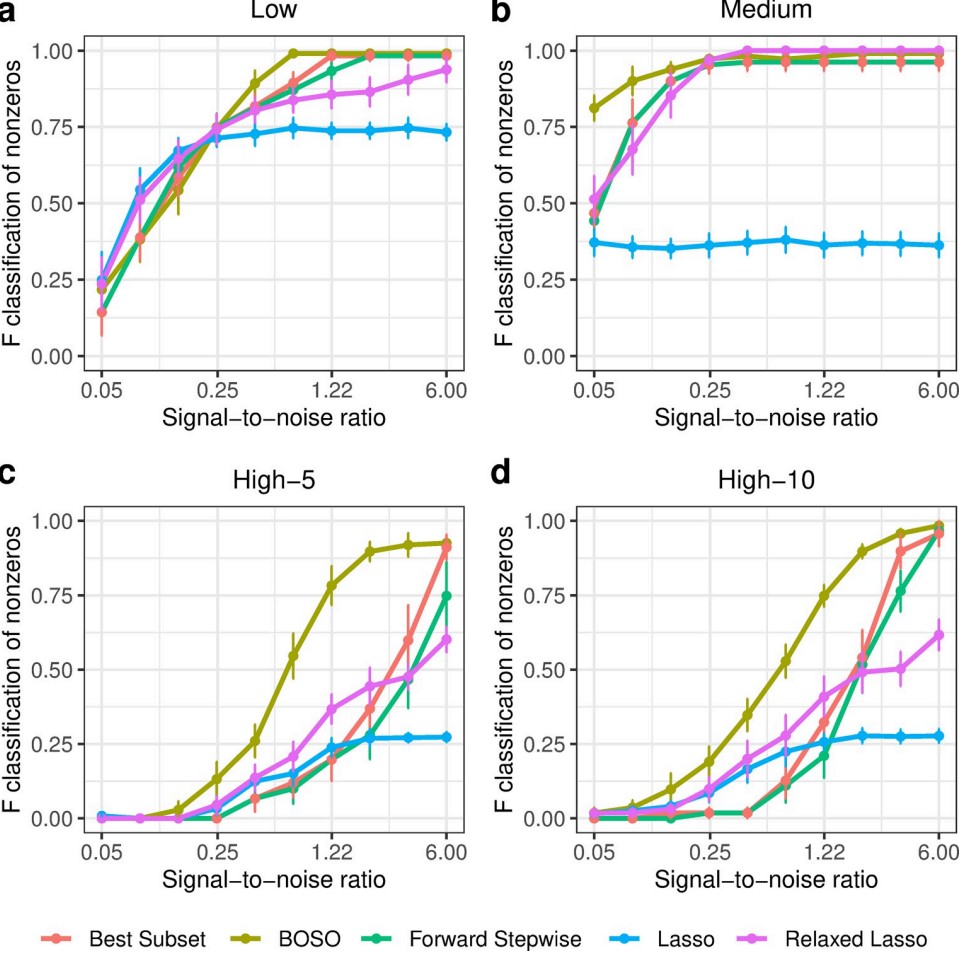

**Fig 2. Performance comparison of BOSO with different feature selection algorithms using F1-score.** a) Low setting; b) Medium setting; c) High-5 setting; d) High-10 setting. Dots and bars represent, respectively, the mean and standard deviation of F1-scores across 10 random samples for the different SNR values.

than Best Subset and Forward Stepwise, and it had mixed outcomes when compared to Lasso and Relaxed Lasso (Fig 2A). For the Medium setting ($p$ = 100, Fig 2B), BOSO and Relaxed Lasso compete to be the most accurate approach, namely BOSO in low SNR values and Relaxed Lasso in high SNR values. Importantly, BOSO achieved the best performance in the High-5 setting (p = 1000, Fig 2C), obtaining more accurate results than the rest of approaches for all the cases. Finally, a similar behavior is observed in the High-10 setting (p = 1000, Fig 2D). According to these results, BOSO is overall more accurate than Best Subset, Forward Stepwise and Lasso and competes with Relaxed Lasso, finding comparable accuracy in low-to-medium-dimensional problems and superior results in high-dimensional scenarios.

In order to gain insights into the type of model obtained from BOSO, in Fig 3 we plotted the number of non-zeros obtained with each method in the simulation presented in Fig 2. It can be seen that BOSO generates a more parsimonious model than Relaxed Lasso and Lasso. This is partially derived from our choice of an information criterion to select the size of the model (in this case eBIC). As a result, BOSO outputs regression models with a significantly lower number of false positives than Lasso and Relaxed Lasso and comparable false negatives (see Figs 4 and 5,

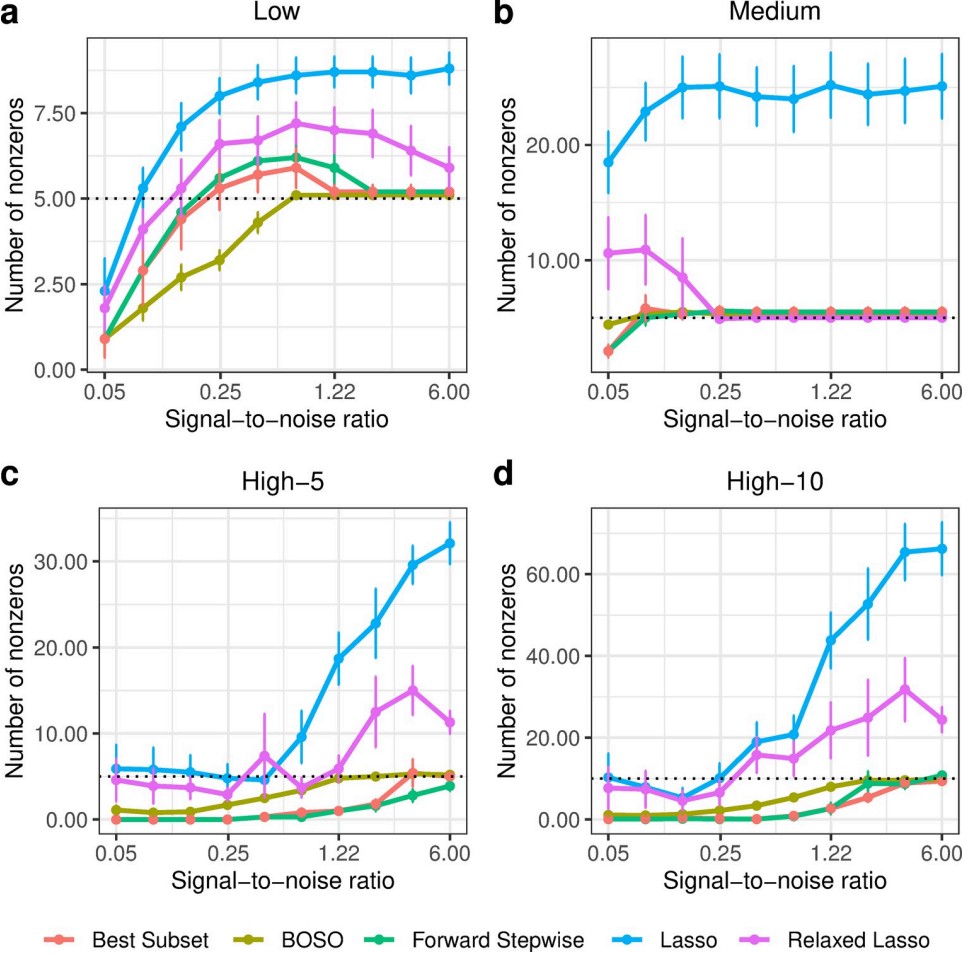

**Fig 3. Performance comparison of BOSO with different feature selection algorithms using Number of non-zeros in the 4 considered problem settings.** a) Low setting; b) Medium setting; c) High-5 setting; d) High-10 setting. Dots and bars represent, respectively, the mean and standard deviation of Number of non-zeros across 10 random samples for different SNR values. The dotted line is the actual value of non-zeros (s) for each SNR value.

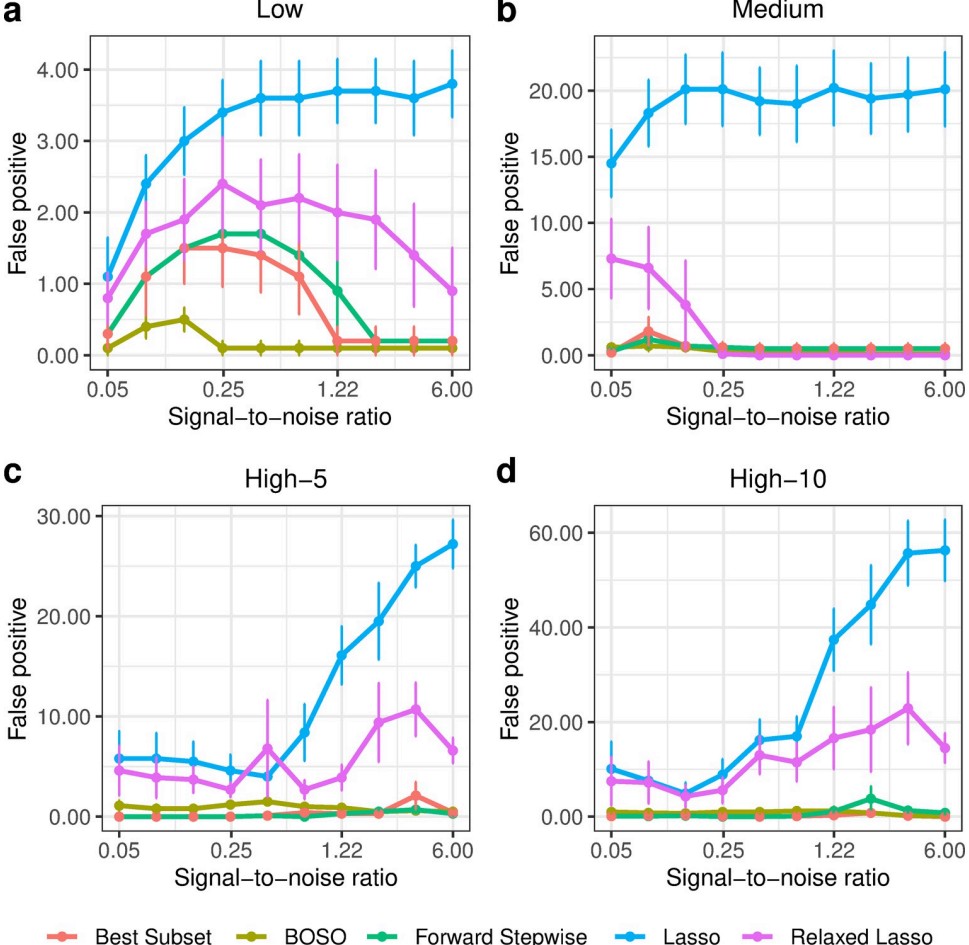

**Fig 4. Performance comparison of BOSO with different feature selection algorithms using False Positives in the 4 considered problem settings.** a) Low setting; b) Medium setting; c) High-5 setting; d) High-10 setting. Dots and bars represent, respectively, the mean and standard deviation of Number of non-zeros across 10 random samples for different SNR values.

respectively). On the other hand, BOSO, Best Subset and Forward Stepwise have similar complexity (Fig 3), but, according to results in Fig 2, Best Subset and Forward Stepwise are less accurate, since they present a higher number of false negatives than BOSO (Fig 5).

A similar behavior is found for beta-type 2 (see S2–S21 Figs), which defines a more complex situation where actual variables contributing to the outcome are correlated with each other. However, we found that BOSO performs worse than Relaxed Lasso for higher correlations in this setting (autocorrelation level 0.70). This is possibly due to the fact that information criterions assume that variables are independent and they are not prepared for cases in which variables present high correlations. This effect is less relevant for more sparse problems, for example, High-5 and Medium.

Results in Figs 2–5 were calculated using eBIC as the information criterion. Fig 6 shows the results presented in Fig 2 for AIC, BIC and eBIC. It can be observed that eBIC and BIC have similar results; in fact, when $p < n$, as in the Low and Medium cases, eBIC is equal to BIC (see Methods section). Differences arise in the case of High-5 and High-10, where eBIC is more

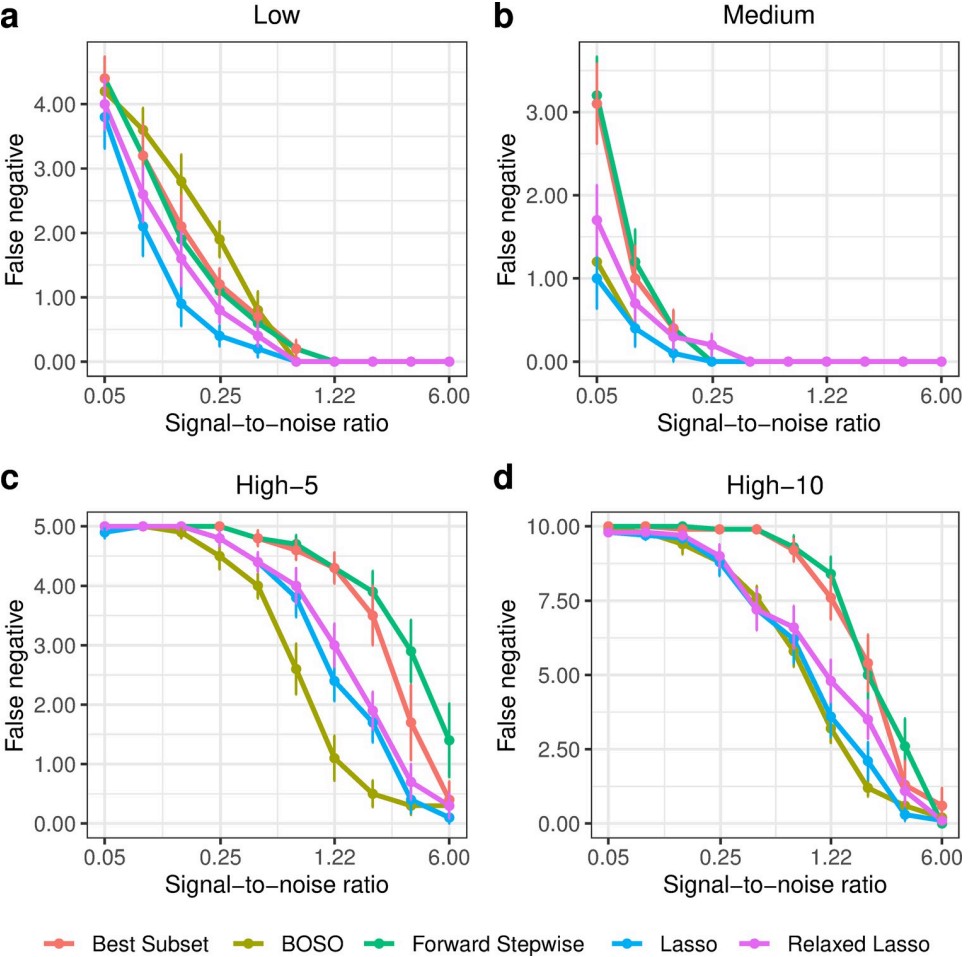

**Fig 5. Performance comparison of BOSO with different feature selection algorithms using False Negatives in the 4 considered problem settings.** a) Low setting; b) Medium setting; c) High-5 setting; d) High-10 setting. Dots and bars represent, respectively, the mean and standard deviation of Number of non-zeros across 10 random samples for different SNR values.

restrictive than BIC, decreasing the number of both false and true positives. This situation is much more extreme in the case of AIC, where the number of false positives is substantially increased with respect to BIC, but it is the one with lowest number of false negatives (see S22–S41 Figs for further details). Although BIC and eBIC present more accurate results than AIC, we considered the 3 information criteria for further analysis.

With respect to computational effort, even using the random block strategy mentioned above, BOSO requires more time than Forward Stepwise, Lasso and Relaxed Lasso. However, BOSO is more efficient than Best Subset and can be run in standard computers, *e.g.* each run in the High-10 setting took us on average 104.6 seconds on a 64 bit Intel(R) Xeon(R) CPU E5-2630 v4 @ 2.20GHz running Linux, setting a maximum of 4 cores and 4 GB of RAM. Further details can be found in S1 Table.

In summary, for feature selection: 1) BOSO shows higher sensitivity than Best Subset and Forward Stepwise; 2) BOSO presents higher specificity than Lasso and Relaxed Lasso; 3) BOSO is a computationally feasible approach in large-sized problems encountered in biomedical research.

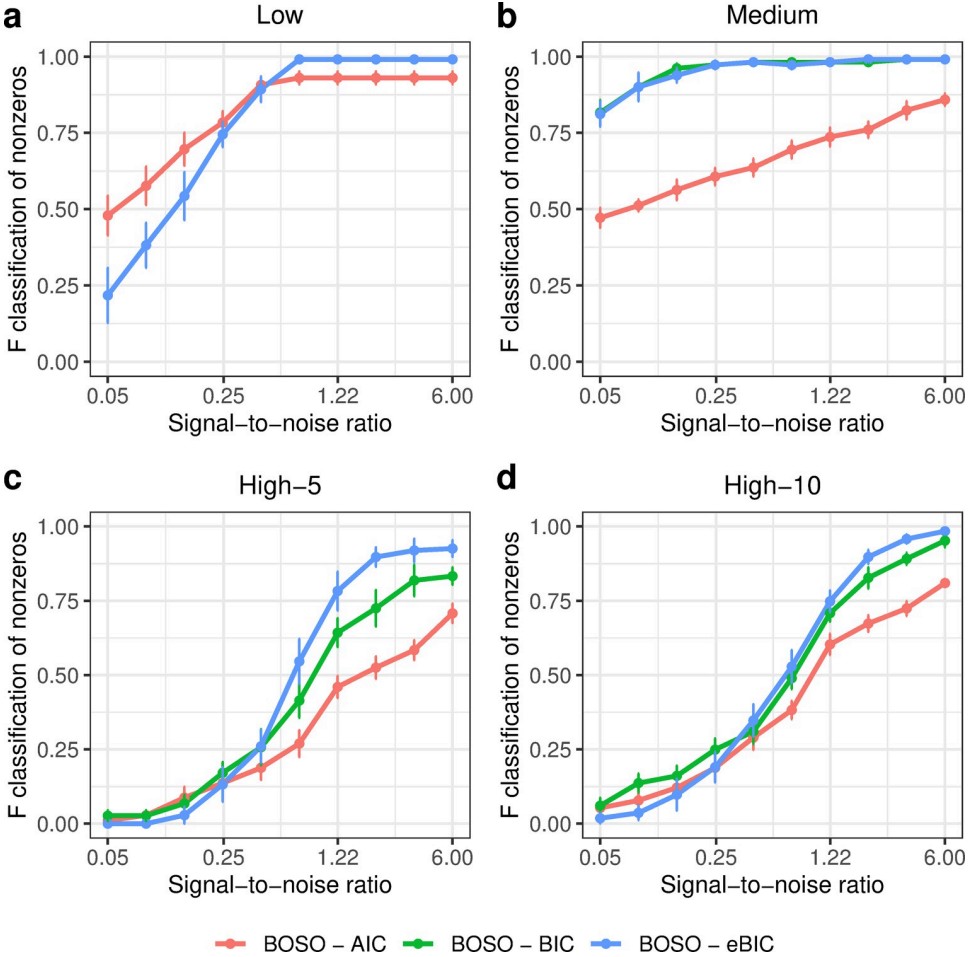

**Fig 6. Performance comparison of BOSO under different information criterions using the F1-score in the 4 considered problem settings.** a) Low setting; b) Medium setting; c) High-5 setting; d) High-10 setting. Dots and bars represent, respectively, the mean and standard deviation of F1-score across 10 random samples for different SNR values. Note here that BOSO-BIC and BOSO-eBIC obtained the same result in the low setting and, for this reason, the blue and green lines overlap in panel a.

## BOSO and drug sensitivity in cancer

We applied BOSO to construct a predictive model of Methotrexate (MTX) cytotoxicity in cancer cell lines. To that end, we used 662 cancer cell lines with the IC50 values of MTX available from the screenings of the GDSC (Genomics of Drug Sensitivity in Cancer) database [28] and RNA-seq data from CCLE (Cancer Cell Line Enyclopedia) [29]. After filtering genes with low mean and variance expression out (see Methods section), we kept 5364 genes (features) as possible predictors of MTX IC50 ($p$ = 5364). In order to guide the learning process, cell lines were randomly grouped into training (40%), validation (40%) and test (20%) sets using the R package *caret* (http://topepo.github.io/caret/index.html) for a homogenous distribution of IC50 values. BOSO was applied to training and validation sets and evaluated with test data in 100 different runs (S2–S4 Tables). We conducted the same analysis with Forward Stepwise, Lasso and Relaxed Lasso (S5–S7 Tables). We excluded Best Subset due to its high computational cost.

From Fig 7A it can be seen that: 1) among different information criteria, the best performance of BOSO in test data was obtained with BIC: mean correlation of 0.612; 2) the models

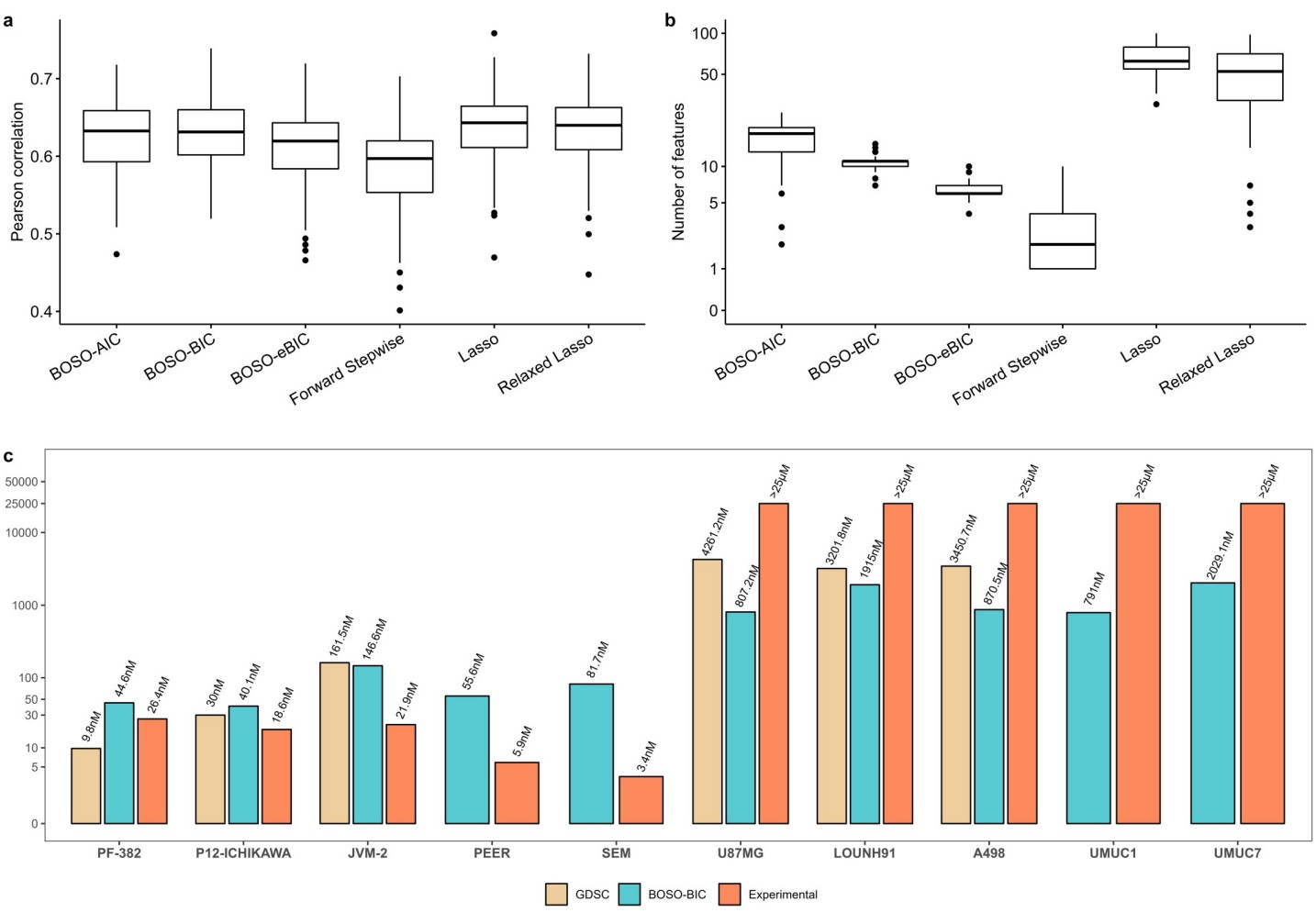

**Fig 7. Prediction of Methotrexate cytotoxicity in cancer.** Using 100 random partitions of data into training, validation and test sets: a) Pearson correlation obtained with BOSO, Forward Stepwise, Lasso and Relaxed in the Test partition; b) Number of active features selected in the approaches included in Fig 7A; c) Experimental validation of IC50 values predicted by the BOSO-BIC algorithm for 5 MTX-sensitive (PF-382, P12-ICHIKAWA, JVM-2, PEER, SEM) and 5 MTX-resistant (U87MG, A498, LOUNH91, UMUC1, UMUC7) cell lines. The cell lines with available GDSC IC50 values (PF-382, P12-ICHIKAWA, JVM-2, U87MG, A498, LOUNH91) were excluded from the model construction process.

derived from Lasso and Relaxed Lasso have similar mean correlation in test data: 0.623 and 0.619, respectively; 3) Forward Stepwise is the least accurate approach (mean correlation of 0.575). On the other hand, there is a striking difference in the number of features: while BOSO and Forward Stepwise predicted on average 10.29 and 2.83, respectively, Lasso and Relaxed Lasso involved more than 56 features (Fig 7B). These results reinforce the conclusions that BOSO generates a more parsimonious model than Lasso and Relaxed Lasso and more accurate model than Forward Stepwise. We repeated the same analysis with 50 drugs available in the GDSC database (S8 Table), finding similar conclusions as the ones obtained for MTX analysis (S42 Fig).

Using the regression models derived by BOSO for the 100 random partitions of training, validation and test data, we predicted the MTX IC50 value for 708 cell lines not included in the GDSC database but with RNA-seq data available in CCLE (S9–S11 Tables). BOSO found clear differences among the distinct cell lines that were considered, with IC50 values ranging from 31.6 nM to 3401 nM. In addition, BOSO predicted a significant difference in the MTX IC50 values for

the top 25% most sensitive and resistant cell lines (Student's t-test p-value = 1.15e-94, S43 Fig for details).

In addition, we conducted *in-vitro* experiments in order to validate our predictive model (see Methods section). First, the IC50 values provided by the GDSC database in 3 MTX-sensitive (PF-382, P12-ICHIKAWA, JVM-2) and 3 MTX-resistant (U87MG, A498, LOUNH91) cell lines (Fig 7C) were validated. This was done because the IC50 values provided by the GDSC database are predicted based on a limited range of experimental screening concentrations[28]. Note here that these 6 cell lines were not used in the model construction process, *i.e.* they were not part of the 662 cell lines used to build the predictive models summarized in Fig 7A and 7B. Second, the IC50 values predicted by BOSO in 2 MTX-sensitive (PEER, SEM) and 2 MTX-resistant (UMUC1, UMUC7) cell lines that were not available in the GDSC database (Fig 7C) were assessed *in-vitro*. Predictions with the rest of methods considered in Fig 7A and 7B can be found in S44 Fig. First, the results predicted from BOSO and GDSC did not present statistically significant differences in the 6 matching cell lines (Student's paired t-test p-value: 0.26). Second, our approach could distinguish between *in-vitro* validated MTX-resistant (n = 5) and MTX-sensitive (n = 5) cell lines (Student's t-test p-value: 4.21e-5). All together indicates that the linear regression model derived by BOSO can be applied to complete the data provided by the GDSC database.

Surprisingly, the most relevant features in BOSO, according to their recurrence in different runs (see S3 Table), are not typically annotated to MTX in drug databases. In particular, the top-5 genes are: LRRC8C, MFNG, RNLS, KBTBD11 and CUEDC1. The individual expression of each gene exhibits a high and significant correlation with MTX IC50 (S45A Fig). Importantly, a linear model with these 5 genes substantially overperforms a model including the 30 genes annotated to MTX in DrugBank (S45B Fig and S12 Table), which shows the relevance of the novel predictors identified.

The importance of these 5 genes in MTX resistance deserves further study and experimentation. However, existing literature provides promising insights about their potential mechanism of action. LRRC8C is a component of the volume-regulated anion channel (VRAC) that has been recently linked to multidrug resistance in cancer in compounds such as cisplatin [30]. MFNG is a manic fringe protein that regulates Notch signaling [31], a pathway previously associated with MTX resistance [32]. KBTBD11 is a tumor suppressor gene that has been identified as differentially expressed in MTX-resistant colon cancer cell lines [33]. CUEDC1 is correlated with estrogen receptor alpha (ERα) [34], which has been found to confer MTX resistance in osteosarcoma cells [35].

## Discussion

The feature selection problem is old in machine learning, but still of high interest to this day. High-dimensional datasets are proliferating in different domains of science and industry, particularly in biomedical research, where high-throughput–omics technologies, mainly DNA-seq and RNA-seq data, are essential tools for biomarker development in the field of personalized medicine and nutrition. In this context, feature selection is a crucial strategy to develop robust machine learning models in problems with limited sample size.

Here, we present BOSO (Bilevel Optimization Selector Operator), a novel feature selection approach for linear regression approaches. BOSO overcomes a complex bilevel optimization problem, linked to the best subset selection problem, based on Mixed-Integer Quadratic Programming. This elegant mathematical transformation is surprisingly novel in the literature. Certainly, existing approaches in the literature address the best subset selection using brute force if possible or heuristic methods for more complex problems [36]. Others do not make use of validation data for feature selection but to select the optimal lenght, as done in Forward

Stepwise. Our strategy is conceptually different and opens new avenues for developing feature selection algorithms in other relevant machine learning tools, such as support vector machines or survival models.

Following the interesting discussion held in the literature [17,18], BOSO was benchmarked with key feature selection algorithms for linear regression models. BOSO falls between Forward Stepwise and Lasso or Relaxed Lasso. Importantly, BOSO shows higher sensitivity than Forward Stepwise and higher specificity than Lasso and Relaxed Lasso in multidimensional problems, which entails a clear advance in machine learning. This improvement is a mixed result of our proposed MIQP and the choice of our information criterion based on BIC. However, we think BOSO could be improved further with information criteria that take into account the correlation between the true variables in the model, as they are currently not prepared for this task.

Proof-of-concept of BOSO was accomplished to predict drug sensitivity in cancer. A detailed analysis was presented for methotrexate (MTX), a well-studied drug targeting cancer metabolism. BOSO showed higher accuracy than Forward Stepwise and derived a more parsimonious model than Lasso and Relaxed Lasso, which reinforces our ability to rule out false positives. This advantage of BOSO is particularly relevant for biomedical applications, since it simplifies the interpretation, validation and posterior exploitation of results (*e.g.* for the development of combinatorial biomarkers). This was illustrated with the 5 most relevant features predicted by BOSO, which exhibits a high predictive power and open new avenues to understand MTX resistance. Finally, we were able to extend the MTX IC50 values provided by the GDSC database to the remaining 708 CCLE cell lines, providing successful experimental validation for 5 MTX-resistant and 5 MTX-sensitive.

In summary, the results here presented illustrate the value of BOSO for the machine learning community and, in particular, for biomedical research, a field where the number of high-dimensional datasets grows at a frenetic pace. We expect to see the application of BOSO to the great variety of methods where Lasso is currently being applied: predictive models of drug sensitivity, resistance or toxicity, construction of gene regulatory networks, biomarker selection, association studies and other relevant questions.

## Methods

### Bilevel optimization in ordinary linear regression

Assume a linear regression model with response vector $y \in R^n$ and design matrix $X \in R^{nx(p+1)}$, where $p$ is the number of predictor variables. The problem of feature selection consists of identifying the subset of predictor variables $Q$ that more accurately predicts the response variable $y$. To address this problem with ordinary linear regression, we split the data into training and validation sets, namely $y = [y^{train}, y^{val}]$ and $X = [X^{train}, X^{val}]$, and construct a standard bilevel quadratic optimization model (Eqs (1)–(4)):

$$\min_Q \ e_Q^{val^T} \cdot e_Q^{val}$$

$$\text{s.t.}$$

Eq (1)

$$y^{val} = X_Q^{val} \cdot \beta_Q + e_Q^{val}$$

Eq (2)

$$\min e_Q^{train^T} \cdot e_Q^{train}$$

$$\text{s.t.}$$

Eq (3)

$$y^{train} = X_Q^{train} \cdot \beta_Q + e_Q^{train}$$

Eq (4)

, where the inner problem (Eqs (3) and (4)) makes use of the training data for a particular subset of features $Q$ ($y^{train}$, $X_Q^{train}$) in order to infer its associated optimal parameters $\beta_Q$ and the outer problem selects the combination of the features $Q$ with the lowest validation (generalization) error. Note here that, in bilevel optimization models, the optimal space of the inner problem is a constraint of the outer problem.

The identification of $Q$ is a combinatorial problem and approaches in the literature follow a heuristic strategy, such as genetic algorithms [37]. We show below that this bilevel quadratic optimization problem can be reformulated as a mixed-integer quadratic programming model, which can be globally solved with standard optimizers such as IBM ILOG CPLEX. Our approach relies on the observation that the optimal solution of the inner problem can be expressed as a set of linear equations that depends on the selected features. Below we detail this transformation step-by-step.

First, let us consider the optimal solutions for the inner problem by assuming that all variables are selected. In that case, following the optimality conditions of ordinary linear regression models (derived from the method of Lagrange multipliers), the inner problem (Eqs (5) and (6)) can be simplified to a linear set of equations (Eq (7)):

$$\min e^{train^T} \cdot e^{train}$$

$$\text{s.t.} \qquad\qquad\qquad\qquad\qquad\qquad\qquad\qquad \text{Eq (5)}$$

$$y^{train} = X^{train} \cdot \beta + e^{train} \qquad\qquad\qquad \text{Eq (6)}$$

$$X^{train^T} \cdot y^{train} = X^{train^T} \cdot X^{train} \cdot \hat{\beta} \qquad\qquad\qquad \text{Eq (7)}$$

In Eq (7), we have one equation for each of the considered features plus the intercept ($p+1$ equations). For the sake of simplicity, by making $a = X^{train^T} \cdot y^{train}$ and $C = X^{train^T} \cdot X^{train}$, where $a \in R^{p+1}$ and $C \in R^{(p+1)x(p+1)}$, we can rewrite the equations algebraically in Eq (8) and uncoupled in Eq (9).

$$a = C \cdot \hat{\beta} \qquad\qquad\qquad\qquad \text{Eq (8)}$$

$$a_i = \sum_{j=1}^{p+1} C_{ij} \cdot \hat{\beta}_j; \; i = 1, \ldots, (p+1) \qquad\qquad\qquad \text{Eq (9)}$$

Importantly, coming back to our initial bilevel quadratic optimization problem, the optimality constraints in Eq (9) only need to be satisfied for the active subset of features $Q$ in the inner problem. In other words, if a feature is not considered in the inner problem, then $\hat{\beta}_j = 0$ but, additionally, its associated constraint in Eq (9) must be neglected. These optimality conditions of the inner problem, which depend on the subset of active variables, can be written as a set of linear equations using binary variables $z_i$, where $z_i = 0$ if a particular feature $i$ is not considered as part of the optimal selection, $z_i = 1$ otherwise. These equations are written in Eqs (10)–(13). Note here that $M$ is a large positive constant.

$$a_i \geq \sum_{j=1}^{p+1} C_{ij} \cdot \hat{\beta}_j - M \cdot (1 - z_i); \; i = 1, \ldots, (p+1) \qquad\qquad \text{Eq (10)}$$

$$a_i \leq \sum_{j=1}^{p+1} C_{ij} \cdot \hat{\beta}_j + M \cdot (1 - z_i); \; i = 1, \ldots, (p+1) \qquad\qquad \text{Eq (11)}$$

$$-M \cdot z_i \leq \hat{\beta}_i \leq M \cdot z_i; \; i = 1, \ldots, (p+1) \qquad \text{Eq (12)}$$

$$z_i = \{0, 1\}; \; i = 1, \ldots, (p+1) \qquad \text{Eq (13)}$$

Now we can re-write the bilevel optimization problem as a single mixed-integer quadratic programming problem (MIQP). Our proposed MIQP directly identifies the subset of features that minimizes the validation error given that their associated parameters β are optimal in the training problem. Full details of our MIQP are detailed in Eqs (14)–(19).

$$min \; e^{val^T} \cdot e^{val} \qquad \text{Eq (14)}$$

s.t.

$$y^{val} = X^{val} \cdot \beta + e^{val} \qquad \text{Eq (15)}$$

$$X^{train^T} \cdot y^{train} \geq X^{train^T} \cdot X^{train} \cdot \beta - M \cdot (1 - z) \qquad \text{Eq (16)}$$

$$X^{train^T} \cdot y^{train} \leq X^{train^T} \cdot X^{train} \cdot \beta + M \cdot (1 - z) \qquad \text{Eq (17)}$$

$$-M \cdot z \leq \beta \leq M \cdot z \qquad \text{Eq (18)}$$

$$z_i = \{0, 1\}; \; i = 1, \ldots, (p+1) \qquad \text{Eq (19)}$$

If this MIQP is applied directly, the resulting solution may suffer from overfitting, particularly in cases where the number of features ($p$) is comparable (or higher) to the number of instances ($n$). To avoid this issue, we iteratively apply this MIQP forcing a specific number of features $K$ ($K = 1,..,p$), as shown in Eq (20), until a specific information criterion (such as AIC, BIC or eBIC) is not further improved.

$$\sum_{j=1}^{p+1} z_j = K \qquad \text{Eq (20)}$$

## Bilevel optimization in ridge regression

Similar to ordinary linear regression, the bilevel optimization model associated with Ridge regression is the following:

$$\min_Q \; e_Q^{val^T} \cdot e_Q^{val} \qquad \text{Eq (21)}$$

s.t.

$$y^{val} = X_Q^{val} \cdot \beta_Q + e_Q^{val} \qquad \text{Eq (22)}$$

$$\min \; e_Q^{train^T} \cdot e_Q^{train} + \delta \cdot \beta_Q^T \cdot \beta_Q \qquad \text{Eq (23)}$$

s.t.

$$y^{train} = X_Q^{train} \cdot \beta_Q + e_Q^{train} \qquad \text{Eq (24)}$$

, where $\delta$ is the regularization parameter.

In this case, when all variables are selected, the optimal solution of the inner problem satisfies the following equation (derived from the method of Lagrange multipliers):

$$X^{train^T} \cdot y^{train} = X^{train^T} \cdot X^{train} \cdot \hat{\beta} + \delta \cdot \hat{\beta} \qquad \text{Eq (25)}$$

With respect to Eq (7) in ordinary linear regression, we added the non-linear term $\delta \cdot \hat{\beta}$. However, for a finite number of $\delta$ values ($\delta_1, \ldots, \delta_m$), as typically used in regularization techniques, we can make it linear through binary variables:

$$X^{train^T} \cdot y^{train} = X^{train^T} \cdot X^{train} \cdot \hat{\beta} + v \qquad \text{Eq (26)}$$

$$v \geq \delta_t \cdot \hat{\beta} - M \cdot (1 - y_t); \ t = 1, \ldots, m \qquad \text{Eq (27)}$$

$$v \leq \delta_t \cdot \hat{\beta} + M \cdot (1 - y_t); \ t = 1, \ldots, m \qquad \text{Eq (28)}$$

$$\sum_{t=1}^{m} y_t = 1 \qquad \text{Eq (29)}$$

Using $y$ variables, we can select the value of $\delta$ and $v$; in particular, when $y_t = 1$, then $v = \delta_t \cdot \hat{\beta}$; when $y_t = 0$, the value of $v$ is not restricted. As shown in Eq (29), we can only have one $y$ variable as active.

Finally, we can amend Eq (26) to take into account feature selection. In a similar way as done above for ordinary linear regression, we obtain again a mixed-integer quadratic programming problem that is summarized below:

$$min \ e^{val^T} \cdot e^{val}$$
$$\text{s.t.} \qquad \text{Eq (30)}$$

$$y^{val} = X^{val} \cdot \beta + e^{val} \qquad \text{Eq (31)}$$

$$X^{train^T} \cdot y^{train} \geq X^{train^T} \cdot X^{train} \cdot \beta + v - M \cdot (1 - z) \qquad \text{Eq (32)}$$

$$X^{train^T} \cdot y^{train} \leq X^{train^T} \cdot X^{train} \cdot \beta + v + M \cdot (1 - z) \qquad \text{Eq (33)}$$

$$-M \cdot z \leq \beta \leq M \cdot z \qquad \text{Eq (34)}$$

$$v \geq \delta_t \cdot \beta - M \cdot (1 - y_t); \ t = 1, \ldots, m \qquad \text{Eq (35)}$$

$$v \leq \delta_t \cdot \beta + M \cdot (1 - y_t); \ t = 1, \ldots, m \qquad \text{Eq (36)}$$

$$\sum_{t=1}^{m} y_t = 1 \qquad \text{Eq (37)}$$

$$\sum_{j=1}^{p+1} z_j = K \qquad \text{Eq (38)}$$

$$z_i = \{0, 1\}; \ i = 1, \ldots, (p + 1) \qquad \text{Eq (39)}$$

$$y_t = \{0, 1\}; \ t = 1, \ldots, m \qquad \text{Eq (40)}$$

As noted above, we iteratively apply this MIQP, Eqs (30)–(40), forcing a specific number of features $K$ ($K = 1,..,p$) until an information criterion is not further improved (see next sub-section). With this approach, we obtain the optimal subset of features $Q$ and the optimal value of the regularization parameter $\delta$. This was the approach used in the Results section. The choice of Ridge regression in the inner layer over ordinary linear regression was done to reduce the variance of the derived model in the event of multicollinearity (high correlation between input variables).

## Extended Bayesian information criterion

eBIC is an extension of BIC (Bayesian Information Criterion) for high-dimensional datasets where $p > n$. For ordinary linear regression, eBIC is defined in Chen and Chen, 2008 [23], as follows:

$$eBIC = n \cdot \log(MSE) + K \cdot \log(n) + 2 \cdot g \cdot \log\binom{p}{K}$$

Eq (41)

, where $n$ is the number of instances, $MSE$ is the Mean Square Error of the regression model for selected features using both training and validation data, $K$ is the number of selected features and $p$ is the total number of features. Note here that $g$ is a consistency parameter. We used the standard value $g = 0.5$ if $p > n$; if $p \leq n$, we fixed $g = 0$, which is equivalent to the Bayesian Information Criterion (BIC). Note here that in the Akaike Information Criterion (AIC), we have $g = 0$ and substitute $log(n)$ by $2$.

Here, we modify the standard eBIC to consider the use of Ridge regression instead of ordinary linear regression. This was done by substituting the number of features $K$ by the effective number of parameters in the model $K_{eff}$ and degrees of freedom ($df(\delta)$):

$$eBIC = n \cdot \log(MSE) + df(\delta) \cdot \log(n) + 2 \cdot g \cdot \log\binom{p}{K_{eff}}$$

Eq (42)

The number of degrees of freedom in Ridge regression is well-known [38]:

$$df(\delta) = trace(X_{Q(K)} \cdot (X_{Q(K)}^T \cdot \delta X_{Q(K)} + \delta \cdot I_K)^{-1} \cdot X_{Q(K)}^T)$$

Eq (43)

, where $X_{Q(K)}$ is the sub-matrix of $X$ only including the columns of the $K$ features selected. Note here that if there is no regularization ($\delta = 0$), the number of effective parameters is precisely $K$. As $df(\delta)$ will be typically non-integer, we round up $K_{eff}$ to the nearest integer:

$$K_{eff} = \min x : \{x \geq df(\delta), x \in Z^+\}$$

Eq (44)

## Computational implementation

In cases with a high number of features, we divide the full set of features into random blocks of features of length $L$ (here $L = 10$) and apply our MIQP approach described above to each block using $m$ different $\delta$ values (here $m = 10$). The selected features in each block are integrated and again divided into random blocks. Our MIQP approach is then applied to each new block. This process is repeated until convergence, namely when the subset of selected features is the same after several iterations or the number of features is less than $L$. In the case of eBIC, in a first stage, in order to select the number of features in each random block, we used BIC, which is a less restrictive strategy. In a second stage, with the resulting subset of features obtained in the first stage, our random block strategy was repeated using a higher $m$ value ($m = 50$ for *low* settings, $m = 100$ for the rest) and eBIC for feature selection. Note here that the minimum and maximum $\delta$ values were extracted from the *glmnet* package [39]. In particular, they correspond to the minimum and maximum value of the lambda parameter involved in the Lasso and

Ridge regression, respectively. Then, the rest of δ values are equally spaced between the minimum and maximum value in a logarithmic scale.

We used IBM ILOG CPLEX to solve the MIQP defined by Eqs (30)–(40). In order to overcome numerical issues derived from the use of the big $M$ method in Eqs (32)–(36), we implemented indicator constraints available in IBM ILOG CPLEX [40]. The code was implemented in the R package BOSO, available on the Comprehensive R Archive Network (https://cran.r-project.org/web/packages/BOSO/index.html) and on GitHub (https://github.com/lvalcarcel/BOSO). We fixed a time limit for each optimization run of 60 seconds on a 64 bit Intel(R) Xeon(R) CPU E5-2630 v4 @ 2.20GHz running Linux, setting a maximum of 4 cores and 4 GB of RAM.

## Drug sensitivity in cancer

For the drug sensitivity analysis, RNA-seq data for different CCLE cancer cell lines was downloaded from the DepMap (Dependency Map) portal (www.depmap.org)[41]. Gene expression levels are provided in log2(TPM+1). We kept for further analysis those genes with: 1) mean expression value across the cell lines greater than 1 TPM; 2) variance across the cell lines greater than one unit. IC50 values were also taken from the DepMap portal.

## Cell culture

PF-382, P12-ICHIKAWA, JVM-2, A-498, LOUNH91, U-87MG, PEER, and SEM cell lines were obtained from the DSMZ or the American Type Culture Collection (ATCC) and were authenticated by performing an STR (short tandem repeat) allele profile. UMUC1 and UMUC7 lines were provided by Dr. Paramio at CIEMAT (Centro de Investigaciones Energéticas, Medioambientales y Tecnológicas). U-87MG was cultured with DMEM medium and the rest cell lines were maintained in culture in RPMI 1640 medium supplemented with fetal bovine serum at 37˚C in a humid atmosphere containing 5% CO2. Aside from UMUC1 and UMUC7, the rest of cell lines were tested for mycoplasma (MycoAlert Sample Kit, Cambrex).

## Methotrexate treatment and cell proliferation assay

Methotrexate (S1210) was purchased from Selleckchem (Houston, TX), dissolved in DMSO at 10mM and stored at -80˚C.

Cell proliferation was analyzed using the CellTiter 96 Aqueous One Solution Cell Proliferation Assay (Promega, Madison, W). This is a colorimetric method for determining the number of viable cells in proliferation. For the assay, suspension cells were cultured by triplicate at a density of 1x10$^6$ cells/mL in 96-well plates (100.000 cells/well, 100µL/well), except for JVM-2 cell line that was cultured at a density of 0.2x10$^6$ cells/mL (20.000 cells/well, 100µL/well). Adherent cells were obtained from 80–90% confluent flasks and 100 µL of cells were seeded at a density of 2500 cells /well in 96-well plates by triplicate. Before addition of the compounds, adherent cells were allowed to attach to the bottom of the wells for 12 hours. In all cases, only the 60 inner wells were used to avoid any border effects.

After 96 hours of MTX treatment at different doses, plates with suspension cells were centrifuged at 800 g for 10 minutes and medium was removed. The plates with adherent cells were flicked to remove medium. Then, cells were incubated with 100 µL/well of medium and 20 µL/well of CellTiter 96 Aqueous One Solution reagent. After 1–3 hours of incubation at 37˚C, the plates were incubated for 1–4 hours, depending on the cell line at 37˚C in a humidified, 5% CO2 atmosphere. The absorbance was recorded at 490 nm using 96-well plate readers until absorbance of control cells without treatment was around 0.8. The background absorbance was measured in wells with only cell line medium and solution reagent. First, the average of the absorbance from

the control wells was subtracted from all other absorbance values. Data were calculated as the percentage of total absorbance of treated cells/absorbance of non-treated cells. The GI50 values were determined using non-linear regression plots with the GraphPad Prism v5 software.

## Supporting information

**S1 Appendix. Synthetic data generation and accuracy metrics.**
(PDF)

**S1 Table. Computation time in the benchmark with synthetic data.** For each setting (Low, Medium, High-5 and High-10), computation time was averaged across all cases considered (1200 cases: 4 beta-types, 3 autocorrelation levels, 10 SNR values and 10 random repetitions).
(XLSX)

**S2 Table. BOSO-AIC IC50 MTX model.** Details of 100 models generated with BOSO–AIC based on different random partitions of data into training, validation and test. Column 'Variable Name' indicates Gene symbol and ENSEMBL ID. Column 'numTimes' indicates the number of times a gene is repeated in the 100 different models. Columns 'seed_1' to 'seed_100' indicates the coefficients of each variable in each random partition, being the value 0 if it is not active.
(XLSX)

**S3 Table. BOSO-BIC IC50 MTX model.** Details of 100 models generated with BOSO–BIC based on different random partitions of data into training, validation and test. Column 'Variable Name' indicates Gene symbol and ENSEMBL ID. Column 'numTimes' indicates the number of times a gene is repeated in the 100 different models. Columns 'seed_1' to 'seed_100' indicates the coefficients of each variable in each random partition, being the value 0 if it is not active.
(XLSX)

**S4 Table. BOSO-eBIC IC50 MTX model.** Details of 100 models generated with BOSO–eBIC based on different random partitions of data into training, validation and test. Column 'Variable Name' indicates Gene symbol and ENSEMBL ID. Column 'numTimes' indicates the number of times a gene is repeated in the 100 different models. Columns 'seed_1' to 'seed_100' indicates the coefficients of each variable in each random partition, being the value 0 if it is not active.
(XLSX)

**S5 Table. Forward Stepwise IC50 MTX model.** Details of 100 models generated with Forward Stepwise based on different random partitions of data into training, validation and test. Column 'Variable Name' indicates Gene symbol and ENSEMBL ID. Column 'numTimes' indicates the number of times a gene is repeated in the 100 different models. Columns 'seed_1' to 'seed_100' indicates the coefficients of each variable in each random partition, being the value 0 if it is not active.
(XLSX)

**S6 Table. Lasso IC50 MTX model.** Details of 100 models generated with Lasso based on different random partitions of data into training, validation and test. Column 'Variable Name' indicates Gene symbol and ENSEMBL ID. Column 'numTimes' indicates the number of times a gene is repeated in the 100 different models. Columns 'seed_1' to 'seed_100' indicates the coefficients of each variable in each random partition, being the value 0 if it is not active.
(XLSX)

**S7 Table. Relaxed Lasso IC50 MTX model.** Details of 100 models generated with Relaxed Lasso based on different random partitions of data into training, validation and test. Column 'Variable Name' indicates Gene symbol and ENSEMBL ID. Column 'numTimes' indicates the number of times a gene is repeated in the 100 different models. Columns 'seed_1' to 'seed_100' indicates the coefficients of each variable in each random partition, being the value 0 if it is not active.
(XLSX)

**S8 Table. Details of 50 drugs in the GDSC database used to compare different feature selection algorithms.** Columns indicate the Drug Name and the number of cell lines for which the IC50 is available in the GDSC database.
(XLSX)

**S9 Table. MTX IC50 (uM) prediction with BOSO-AIC model.** Using the BOSO-AIC model, prediction of MTX IC50 values in micro molar for cell lines that are not included in the training process: 708 cell lines that are not present in the GDSC database and 6 cell lines part of the GDSC database for experimental validation purposes. Column 'DepMap_ID' is the identifier of the cell line in the DepMap initiative; 'stripped_cell_line_name' is the name of the cell line in computer-friendly language; 'CCLE_Name' is the name of the cell line and corresponding tissue; 'Mean' is the mean prediction across 100 different runs; 'Seed_1'–'Seed_100' represent the prediction of MTX IC50 value for each cell line.
(XLSX)

**S10 Table. MTX IC50 (uM) prediction with BOSO-BIC model.** Using the BOSO-BIC model, prediction of MTX IC50 values in micro molar for cell lines that are not included in the training process: 708 cell lines that are not present in the GDSC database and 6 cell lines part of the GDSC database for experimental validation purposes. Column 'DepMap_ID' is the identifier of the cell line in the DepMap initiative; 'stripped_cell_line_name' is the name of the cell line in computer-friendly language; 'CCLE_Name' is the name of the cell line and corresponding tissue; 'Mean' is the mean prediction across 100 different runs; 'Seed_1'–'Seed_100' represent the prediction of MTX IC50 value for each cell line.
(XLSX)

**S11 Table. MTX IC50 (uM) prediction with BOSO-eBIC model.** Using the BOSO-eBIC model, prediction of MTX IC50 values in micro molar for cell lines that are not included in the training process: 708 cell lines not present in the GDSC database and 6 cell lines part of the GDSC database for experimental validation purposes. Column 'DepMap_ID' is the identifier of the cell line in the DepMap initiative; 'stripped_cell_line_name' is the name of the cell line in computer-friendly language; 'CCLE_Name' is the name of the cell line and corresponding tissue; 'Mean' is the mean prediction across 100 different runs; 'Seed_1'–'Seed_100' represent the prediction of MTX IC50 value for each cell line.
(XLSX)

**S12 Table. DrugBank genes annotated to Methotrexate (MTX).** Genes annotated to Methotrexate in DrugBank. The type of interaction is described in the column Mechanism.
(XLSX)

**S1 Fig. Illustration of the random block strategy implemented in the BOSO algorithm.** An example dataset with 7 features is split into training and validation sets. We defined random blocks of features of size L = 3. Green boxes represent the optimal selected features for a specific K value in certain block. In the first iteration, the dataset is separated in {X5, X7, X2}, {X1, X4, X3} and {X6}. Applying the BOSO algorithm to each block, we selected {X5, X2} in the

first block, {X4, X3} in the secondo block and {X6} in the third block. Resulting variables are resampled again and randomly distributed into different blocks. In the second iteration, the blocks are {X2, X6, X4} and {X3, X5}. After BOSO, there are three remaining variables {X2, X6, X3}, which equals the block size. The final problem is re-solved, resulting in the optimal feature selection, which is {X3, X6}
(TIF)

**S2 Fig. F statistic in the Low setting.** This accuracy metric is presented for the different feature selection methods (Best Subset, BOSO, Forward Stepwise, Lasso and Relaxed Lasso) and scenarios (according to Beta-type, autocorrelation levels and signal-to-noise ratio (SNR) levels) considered in the main text. S1 Appendix provides full details of the different situations considered. Points and error bars represent the mean and standard deviation in 10 random simulations, respectively. Note here that n is the number of instances, p is the total available features and s is the actual number of features contributing to the response variable.
(TIF)

**S3 Fig. Number of non-zero coefficients in the Low setting.** This accuracy metric is presented for the different feature selection methods (Best Subset, BOSO, Forward Stepwise, Lasso and Relaxed Lasso) and scenarios (according to Beta-type, autocorrelation levels and signal-to-noise ratio (SNR) levels) considered in the main text. S1 Appendix provides full details of the different situations considered. Points and error bars represent the mean and standard deviation in 10 random simulations, respectively. Note here that n is the number of instances, p is the total available features and s is the actual number of features contributing to the response variable. Dotted line represents the actual number of features.
(TIF)

**S4 Fig. False Positives in the Low setting.** This accuracy metric is presented for the different feature selection methods (Best Subset, BOSO, Forward Stepwise, Lasso and Relaxed Lasso) and scenarios (according to Beta-type, autocorrelation levels and signal-to-noise ratio (SNR) levels) considered in the main text. S1 Appendix provides full details of the different situations considered. Points and error bars represent the mean and standard deviation in 10 random simulations, respectively. Note here that n is the number of instances, p is the total available features and s is the actual number of features contributing to the response variable.
(TIF)

**S5 Fig. False Negatives in the Low setting.** This accuracy metric is presented for the different feature selection methods (Best Subset, BOSO, Forward Stepwise, Lasso and Relaxed Lasso) and scenarios (according to Beta-type, autocorrelation levels and signal-to-noise ratio (SNR) levels) considered in the main text. S1 Appendix provides full details of the different situations considered. Points and error bars represent the mean and standard deviation in 10 random simulations, respectively. Note here that n is the number of instances, p is the total available features and s is the actual number of features contributing to the response variable.
(TIF)

**S6 Fig. Relative Test Error in the Low setting.** This accuracy metric is presented for the different feature selection methods (Best Subset, BOSO, Forward Stepwise, Lasso and Relaxed Lasso) and scenarios (according to Beta-type, autocorrelation levels and signal-to-noise ratio (SNR) levels) considered in the main text. S1 Appendix provides full details of the different situations considered. Points and error bars represent the mean and standard deviation in 10 random simulations, respectively. Note here that n is the number of instances, p is the total available features and s is the actual number of features contributing to the response variable.

Dotted curve represents the results for the null model.
(TIF)

**S7 Fig. F statistic in the Medium setting.** This accuracy metric is presented for the different feature selection methods (Best Subset, BOSO, Forward Stepwise, Lasso and Relaxed Lasso) and scenarios (according to Beta-type, autocorrelation levels and signal-to-noise ratio (SNR) levels) considered in the main text. S1 Appendix provides full details of the different situations considered. Points and error bars represent the mean and standard deviation in 10 random simulations, respectively. Note here that n is the number of instances, p is the total available features and s is the actual number of features contributing to the response variable.
(TIF)

**S8 Fig. Number of non-zero coefficients in the Medium setting.** This accuracy metric is presented for the different feature selection methods (Best Subset, BOSO, Forward Stepwise, Lasso and Relaxed Lasso) and scenarios (according to Beta-type, autocorrelation levels and signal-to-noise ratio (SNR) levels) considered in the main text. S1 Appendix provides full details of the different situations considered. Points and error bars represent the mean and standard deviation in 10 random simulations, respectively. Note here that n is the number of instances, p is the total available features and s is the actual number of features contributing to the response variable. Dotted line represents the actual number of features.
(TIF)

**S9 Fig. False Positives in the Medium setting.** This accuracy metric is presented for the different feature selection methods (Best Subset, BOSO, Forward Stepwise, Lasso and Relaxed Lasso) and scenarios (according to Beta-type, autocorrelation levels and signal-to-noise ratio (SNR) levels) considered in the main text. S1 Appendix provides full details of the different situations considered. Points and error bars represent the mean and standard deviation in 10 random simulations, respectively. Note here that n is the number of instances, p is the total available features and s is the actual number of features contributing to the response variable.
(TIF)

**S10 Fig. False Negatives in the Medium setting.** This accuracy metric is presented for the different feature selection methods (Best Subset, BOSO, Forward Stepwise, Lasso and Relaxed Lasso) and scenarios (according to Beta-type, autocorrelation levels and signal-to-noise ratio (SNR) levels) considered in the main text. S1 Appendix provides full details of the different situations considered. Points and error bars represent the mean and standard deviation in 10 random simulations, respectively. Note here that n is the number of instances, p is the total available features and s is the actual number of features contributing to the response variable.
(TIF)

**S11 Fig. Relative Test Error in the Medium setting.** This accuracy metric is presented for the different feature selection methods (Best Subset, BOSO, Forward Stepwise, Lasso and Relaxed Lasso) and scenarios (according to Beta-type, autocorrelation levels and signal-to-noise ratio (SNR) levels) considered in the main text. S1 Appendix provides full details of the different situations considered. Points and error bars represent the mean and standard deviation in 10 random simulations, respectively. Note here that n is the number of instances, p is the total available features and s is the actual number of features contributing to the response variable. Dotted line represents the results for the null model.
(TIF)

**S12 Fig. F statistic in the High-5 setting.** This accuracy metric is presented for the different feature selection methods (Best Subset, BOSO, Forward Stepwise, Lasso and Relaxed Lasso)

and scenarios (according to Beta-type, autocorrelation levels and signal-to-noise ratio (SNR) levels) considered in the main text. S1 Appendix provides full details of the different situations considered. Points and error bars represent the mean and standard deviation in 10 random simulations, respectively. Note here that n is the number of instances, p is the total available features and s is the actual number of features contributing to the response variable.
(TIF)

**S13 Fig. Number of non-zero coefficients in the High-5 setting.** This accuracy metric is presented for the different feature selection methods (Best Subset, BOSO, Forward Stepwise, Lasso and Relaxed Lasso) and scenarios (according to Beta-type, autocorrelation levels and signal-to-noise ratio (SNR) levels) considered in the main text. S1 Appendix provides full details of the different situations considered. Points and error bars represent the mean and standard deviation in 10 random simulations, respectively. Note here that n is the number of instances, p is the total available features and s is the actual number of features contributing to the response variable. Dotted line represents the actual number of features.
(TIF)

**S14 Fig. False Positives in the High-5 setting.** This accuracy metric is presented for the different feature selection methods (Best Subset, BOSO, Forward Stepwise, Lasso and Relaxed Lasso) and scenarios (according to Beta-type, autocorrelation levels and signal-to-noise ratio (SNR) levels) considered in the main text. S1 Appendix provides full details of the different situations considered. Points and error bars represent the mean and standard deviation in 10 random simulations, respectively. Note here that n is the number of instances, p is the total available features and s is the actual number of features contributing to the response variable.
(TIF)

**S15 Fig. False Negatives in the High-5 setting.** This accuracy metric is presented for the different feature selection methods (Best Subset, BOSO, Forward Stepwise, Lasso and Relaxed Lasso) and scenarios (according to Beta-type, autocorrelation levels and signal-to-noise ratio (SNR) levels) considered in the main text. S1 Appendix provides full details of the different situations considered. Points and error bars represent the mean and standard deviation in 10 random simulations, respectively. Note here that n is the number of instances, p is the total available features and s is the actual number of features contributing to the response variable.
(TIF)

**S16 Fig. Relative Test Error in the High-5 setting.** This accuracy metric is presented for the different feature selection methods (Best Subset, BOSO, Forward Stepwise, Lasso and Relaxed Lasso) and scenarios (according to Beta-type, autocorrelation levels and signal-to-noise ratio (SNR) levels) considered in the main text. S1 Appendix provides full details of the different situations considered. Points and error bars represent the mean and standard deviation in 10 random simulations, respectively. Note here that n is the number of instances, p is the total available features and s is the actual number of features contributing to the response variable. Dotted curve represents the results for the null model.
(TIF)

**S17 Fig. F statistic in the High-10 setting.** This accuracy metric is presented for the different feature selection methods (Best Subset, BOSO, Forward Stepwise, Lasso and Relaxed Lasso) and scenarios (according to Beta-type, autocorrelation levels and signal-to-noise ratio (SNR) levels) considered in the main text. S1 Appendix provides full details of the different situations considered. Points and error bars represent the mean and standard deviation in 10 random simulations, respectively. Note here that n is the number of instances, p is the total available

features and s is the actual number of features contributing to the response variable.
(TIF)

**S18 Fig. Number of non-zero coefficients in the High-10 setting.** This accuracy metric is presented for the different feature selection methods (Best Subset, BOSO, Forward Stepwise, Lasso and Relaxed Lasso) and scenarios (according to Beta-type, autocorrelation levels and signal-to-noise ratio (SNR) levels) considered in the main text. S1 Appendix provides full details of the different situations considered. Points and error bars represent the mean and standard deviation in 10 random simulations, respectively. Note here that n is the number of instances, p is the total available features and s is the actual number of features contributing to the response variable. Dotted line represents the actual number of features.
(TIF)

**S19 Fig. False Positives in the High-10 setting.** This accuracy metric is presented for the different feature selection methods (Best Subset, BOSO, Forward Stepwise, Lasso and Relaxed Lasso) and scenarios (according to Beta-type, autocorrelation levels and signal-to-noise ratio (SNR) levels) considered in the main text. S1 Appendix provides full details of the different situations considered. Points and error bars represent the mean and standard deviation in 10 random simulations, respectively. Note here that n is the number of instances, p is the total available features and s is the actual number of features contributing to the response variable.
(TIF)

**S20 Fig. False Negatives in the High-10 setting.** This accuracy metric is presented for the different feature selection methods (Best Subset, BOSO, Forward Stepwise, Lasso and Relaxed Lasso) and scenarios (according to Beta-type, autocorrelation levels and signal-to-noise ratio (SNR) levels) considered in the main text. S1 Appendix provides full details of the different situations considered. Points and error bars represent the mean and standard deviation in 10 random simulations, respectively. Note here that n is the number of instances, p is the total available features and s is the actual number of features contributing to the response variable.
(TIF)

**S21 Fig. Relative Test Error in the High-10 setting.** This accuracy metric is presented for the different feature selection methods (Best Subset, BOSO, Forward Stepwise, Lasso and Relaxed Lasso) and scenarios (according to Beta-type, autocorrelation levels and signal-to-noise ratio (SNR) levels) considered in the main text. S1 Appendix provides full details of the different situations considered. Points and error bars represent the mean and standard deviation in 10 random simulations, respectively. Note here that n is the number of instances, p is the total available features and s is the actual number of features contributing to the response variable. Dotted curve represents the results for the null model.
(TIF)

**S22 Fig. F statistic in the Low setting for BOSO under different information criteria.** This accuracy metric is presented for BOSO under different information criteria (BOSO—AIC, BOSO—BIC and BOSO—eBIC) and scenarios (according to Beta-type, autocorrelation levels and signal-to-noise ratio (SNR) levels) considered in the main text. S1 Appendix provides full details of the different situations considered. Points and error bars represent the mean and standard deviation in 10 random simulations, respectively. Note here that n is the number of instances, p is the total available features and s is the actual number of features contributing to the response variable.
(TIF)

**S23 Fig. Number of non-zero coefficients in the Low setting for BOSO under different information criteria.** This accuracy metric is presented for BOSO under different information

criteria (BOSO—AIC, BOSO—BIC and BOSO—eBIC) and scenarios (according to Beta-type, autocorrelation levels and signal-to-noise ratio (SNR) levels) considered in the main text. S1 Appendix provides full details of the different situations considered. Points and error bars represent the mean and standard deviation in 10 random simulations, respectively. Note here that n is the number of instances, p is the total available features and s is the actual number of features contributing to the response variable. The dotted line is the actual number of features. (TIF)

**S24 Fig. False Positives in the Low setting for BOSO under different information criteria.** This accuracy metric is presented for BOSO under different information criteria (BOSO—AIC, BOSO—BIC and BOSO—eBIC) and scenarios (according to Beta-type, autocorrelation levels and signal-to-noise ratio (SNR) levels) considered in the main text. S1 Appendix provides full details of the different situations considered. Points and error bars represent the mean and standard deviation in 10 random simulations, respectively. Note here that n is the number of instances, p is the total available features and s is the actual number of features contributing to the response variable. (TIF)

**S25 Fig. False Negatives in the Low setting for BOSO under different information criteria.** This accuracy metric is presented for BOSO under different information criteria (BOSO—AIC, BOSO—BIC and BOSO—eBIC) and scenarios (according to Beta-type, autocorrelation levels and signal-to-noise ratio (SNR) levels) considered in the main text. S1 Appendix provides full details of the different situations considered. Points and error bars represent the mean and standard deviation in 10 random simulations, respectively. Note here that n is the number of instances, p is the total available features and s is the actual number of features contributing to the response variable. (TIF)

**S26 Fig. Relative Test Error in the Low setting for BOSO under different information criteria.** This accuracy metric is presented for BOSO under different information criteria (BOSO—AIC, BOSO—BIC and BOSO—eBIC) and scenarios (according to Beta-type, autocorrelation levels and signal-to-noise ratio (SNR) levels) considered in the main text. S1 Appendix provides full details of the different situations considered. Points and error bars represent the mean and standard deviation in 10 random simulations, respectively. Note here that n is the number of instances, p is the total available features and s is the actual number of features contributing to the response variable. Dotted curve represents the results for the null model. (TIF)

**S27 Fig. F statistic in the Medium setting for BOSO under different information criteria.** This accuracy metric is presented for BOSO under different information criteria (BOSO—AIC, BOSO—BIC and BOSO—eBIC) and scenarios (according to Beta-type, autocorrelation levels and signal-to-noise ratio (SNR) levels) considered in the main text. S1 Appendix provides full details of the different situations considered. Points and error bars represent the mean and standard deviation in 10 random simulations, respectively. Note here that n is the number of instances, p is the total available features and s is the actual number of features contributing to the response variable. (TIF)

**S28 Fig. Number of non-zero coefficients in the Medium setting for BOSO under different information criteria.** This accuracy metric is presented for BOSO under different information criteria (BOSO—AIC, BOSO—BIC and BOSO—eBIC) and scenarios (according to Beta-type,

autocorrelation levels and signal-to-noise ratio (SNR) levels) considered in the main text. S1 Appendix provides full details of the different situations considered. Points and error bars represent the mean and standard deviation in 10 random simulations, respectively. Note here that n is the number of instances, p is the total available features and s is the actual number of features contributing to the response variable. Dotted line represents the actual number of features.
(TIF)

**S29 Fig. False Positives in the Medium setting for BOSO under different information criteria.** This accuracy metric is presented for BOSO under different information criteria (BOSO—AIC, BOSO—BIC and BOSO—eBIC) and scenarios (according to Beta-type, autocorrelation levels and signal-to-noise ratio (SNR) levels) considered in the main text. S1 Appendix provides full details of the different situations considered. Points and error bars represent the mean and standard deviation in 10 random simulations, respectively. Note here that n is the number of instances, p is the total available features and s is the actual number of features contributing to the response variable.
(TIF)

**S30 Fig. False Negatives in the Medium setting for BOSO under different information criteria.** This accuracy metric is presented for BOSO under different information criteria (BOSO—AIC, BOSO—BIC and BOSO—eBIC) and scenarios (according to Beta-type, autocorrelation levels and signal-to-noise ratio (SNR) levels) considered in the main text. S1 Appendix provides full details of the different situations considered. Points and error bars represent the mean and standard deviation in 10 random simulations, respectively. Note here that n is the number of instances, p is the total available features and s is the actual number of features contributing to the response variable.
(TIF)

**S31 Fig. Relative Test Error in the Medium setting for BOSO under different information criteria.** This accuracy metric is presented for BOSO under different information criteria (BOSO—AIC, BOSO—BIC and BOSO—eBIC) and scenarios (according to Beta-type, autocorrelation levels and signal-to-noise ratio (SNR) levels) considered in the main text. S1 Appendix provides full details of the different situations considered. Points and error bars represent the mean and standard deviation in 10 random simulations, respectively. Note here that n is the number of instances, p is the total available features and s is the actual number of features contributing to the response variable. Dotted line represents the results for the null model.
(TIF)

**S32 Fig. F statistic in the High-5 setting for BOSO under different information criteria.** This accuracy metric is presented for BOSO under different information criteria (BOSO—AIC, BOSO—BIC and BOSO—eBIC) and scenarios (according to Beta-type, autocorrelation levels and signal-to-noise ratio (SNR) levels) considered in the main text. S1 Appendix provides full details of the different situations considered. Points and error bars represent the mean and standard deviation in 10 random simulations, respectively. Note here that n is the number of instances, p is the total available features and s is the actual number of features contributing to the response variable.
(TIF)

**S33 Fig. Number of non-zero coefficients in the High-5 setting for BOSO under different information criteria.** This accuracy metric is presented for BOSO under different information

criteria (BOSO—AIC, BOSO—BIC and BOSO—eBIC) and scenarios (according to Beta-type, autocorrelation levels and signal-to-noise ratio (SNR) levels) considered in the main text. S1 Appendix provides full details of the different situations considered. Points and error bars represent the mean and standard deviation in 10 random simulations, respectively. Note here that n is the number of instances, p is the total available features and s is the actual number of features contributing to the response variable. The dotted line represents the actual number of features. (TIF)

**S34 Fig. False Positives in the High-5 setting for BOSO under different information criteria.** This accuracy metric is presented for BOSO under different information criteria (BOSO—AIC, BOSO—BIC and BOSO—eBIC) and scenarios (according to Beta-type, autocorrelation levels and signal-to-noise ratio (SNR) levels) considered in the main text. S1 Appendix provides full details of the different situations considered. Points and error bars represent the mean and standard deviation in 10 random simulations, respectively. Note here that n is the number of instances, p is the total available features and s is the actual number of features contributing to the response variable. (TIF)

**S35 Fig. False Negatives in the High-5 setting for BOSO under different information criteria.** This accuracy metric is presented for BOSO under different information criteria (BOSO—AIC, BOSO—BIC and BOSO—eBIC) and scenarios (according to Beta-type, autocorrelation levels and signal-to-noise ratio (SNR) levels) considered in the main text. S1 Appendix provides full details of the different situations considered. Points and error bars represent the mean and standard deviation in 10 random simulations, respectively. Note here that n is the number of instances, p is the total available features and s is the actual number of features contributing to the response variable. (TIF)

**S36 Fig. Relative Test Error in the High-5 setting for BOSO under different information criteria.** This accuracy metric is presented for BOSO under different information criteria (BOSO—AIC, BOSO—BIC and BOSO—eBIC) and scenarios (according to Beta-type, autocorrelation levels and signal-to-noise ratio (SNR) levels) considered in the main text. S1 Appendix provides full details of the different situations considered. Points and error bars represent the mean and standard deviation in 10 random simulations, respectively. Note here that n is the number of instances, p is the total available features and s is the actual number of features contributing to the response variable. Dotted curve represents the results for the null model. (TIF)

**S37 Fig. F statistic in the High-10 setting for BOSO under different information criteria.** This accuracy metric is presented for BOSO under different information criteria (BOSO—AIC, BOSO—BIC and BOSO—eBIC) and scenarios (according to Beta-type, autocorrelation levels and signal-to-noise ratio (SNR) levels) considered in the main text. S1 Appendix provides full details of the different situations considered. Points and error bars represent the mean and standard deviation in 10 random simulations, respectively. Note here that n is the number of instances, p is the total available features and s is the actual number of features contributing to the response variable. (TIF)

**S38 Fig. Number of non-zero coefficients in the High-10 setting for BOSO under different information criteria.** This accuracy metric is presented for BOSO under different information

criteria (BOSO—AIC, BOSO—BIC and BOSO—eBIC) and scenarios (according to Beta-type, autocorrelation levels and signal-to-noise ratio (SNR) levels) considered in the main text. S1 Appendix provides full details of the different situations considered. Points and error bars represent the mean and standard deviation in 10 random simulations, respectively. Note here that n is the number of instances, p is the total available features and s is the actual number of features contributing to the response variable. The dotted line represents the actual number of features.
(TIF)

**S39 Fig. False Positives in the High-10 setting for BOSO under different information criteria.** This accuracy metric is presented for BOSO under different information criteria (BOSO—AIC, BOSO—BIC and BOSO—eBIC) and scenarios (according to Beta-type, autocorrelation levels and signal-to-noise ratio (SNR) levels) considered in the main text. S1 Appendix provides full details of the different situations considered. Points and error bars represent the mean and standard deviation in 10 random simulations, respectively. Note here that n is the number of instances, p is the total available features and s is the actual number of features contributing to the response variable.
(TIF)

**S40 Fig. False Negatives in the High-10 setting for BOSO under different information criteria.** This accuracy metric is presented for BOSO under different information criteria (BOSO—AIC, BOSO—BIC and BOSO—eBIC) and scenarios (according to Beta-type, autocorrelation levels and signal-to-noise ratio (SNR) levels) considered in the main text. S1 Appendix provides full details of the different situations considered. Points and error bars represent the mean and standard deviation in 10 random simulations, respectively. Note here that n is the number of instances, p is the total available features and s is the actual number of features contributing to the response variable.
(TIF)

**S41 Fig. Relative Test Error in the High-10 setting for BOSO under different information criteria.** This accuracy metric is presented for BOSO under different information criteria (BOSO—AIC, BOSO—BIC and BOSO—eBIC) and scenarios (according to Beta-type, autocorrelation levels and signal-to-noise ratio (SNR) levels) considered in the main text. S1 Appendix provides full details of the different situations considered. Points and error bars represent the mean and standard deviation in 10 random simulations, respectively. Note here that n is the number of instances, p is the total available features and s is the actual number of features contributing to the response variable. Dotted curve represents the results for the null model.
(TIF)

**S42 Fig. Prediction of IC50 values for 50 drugs present in the GDSC database.** a) For 20 random partitions into training, validation and test data of the 50 drugs detailed in S8 Table, comparison of the Pearson Correlation values between GDSC IC50 and predicted IC50 values with BOSO-BIC, BOSO-eBIC, Forward Stepwise, Lasso and Relaxed Lasso, respectively, in the Test partition; b) Summary table of mean Pearson Correlation values for the analyzed cases in 'a' panel in the three data partitions; c) Comparison of number of active features for the analyzed cases in 'a'; d) Summary table for the mean number of selected variables for the analyzed cases in 'a'.
(TIF)

**S43 Fig. Comparison of predicted log(IC50[μM]) for the top 25% most sensitive and resistant cell lines with the different methods included in the main text.** IC50 for each cell line

were predicted using the mean value across 100 runs considered in Fig 7. Q1 involves cell lines with a predicted IC50 below the first quartile (sensitive cell lines), whereas Q4 cell lines with a predicted IC50 above the third quartile (resistant cell lines). In order to avoid overfitting, we considered 708 cell lines in CCLE that were not included in the GDSC database.
(TIF)

**S44 Fig. Comparison between experimentally measured IC50 values of MTX and predicted values with different computational methods.** a) BOSO—AIC; b) BOSO—BIC; c) BOSO—eBIC; d) Forward Stepwise; e) Lasso; f) Relaxed Lasso. Predicted values are the mean values obtained with 100 random seeds.
(TIF)

**S45 Fig. Summary of 5 best-ranked features in BOSO and accuracy comparison with features extracted from DrugBank.** a) For each of the 5 best-ranked genes obtained from BOSO (LRRC8C, MFNG, RNLS, KBTBD11, CUEDC1), dot plot showing its corresponding CCLE expression level (x-axis) and MTX IC50 values (y-axis) for cell lines available in the GDSC database. The table shows the Pearson correlation rho value and its associated p-value for each these 5 genes. b) Ridge regression model of MTX IC50 value using as predictors i) genes annotated to MTX in DrugBank (see S12 Table), ii) 5 best-ranked genes obtained from BOSO and iii) the union of both subsets of genes. The table show the correlation between predicted and actual MTX IC50 values for training, validation and test set.
(TIF)

## Author Contributions

**Conceptualization:** Francisco J. Planes.

**Data curation:** Xabier Cendoya.

**Formal analysis:** Luis V. Valcárcel, Xabier Cendoya, Ángel Rubio, Francisco J. Planes.

**Funding acquisition:** Felipe Prósper, Francisco J. Planes.

**Investigation:** Luis V. Valcárcel, Edurne San José-Enériz, Xabier Agirre, Felipe Prósper, Francisco J. Planes.

**Methodology:** Luis V. Valcárcel, Ángel Rubio, Francisco J. Planes.

**Project administration:** Francisco J. Planes.

**Resources:** Edurne San José-Enériz, Xabier Agirre, Felipe Prósper, Francisco J. Planes.

**Software:** Luis V. Valcárcel, Xabier Cendoya, Ángel Rubio.

**Supervision:** Xabier Agirre, Felipe Prósper, Francisco J. Planes.

**Validation:** Edurne San José-Enériz, Xabier Agirre.

**Visualization:** Luis V. Valcárcel.

**Writing – original draft:** Luis V. Valcárcel, Francisco J. Planes.

**Writing – review & editing:** Luis V. Valcárcel, Edurne San José-Enériz, Xabier Cendoya, Ángel Rubio, Xabier Agirre, Felipe Prósper, Francisco J. Planes.

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
