## [Decision Letter · Decision Letter 0]

10 Jan 2022

Dear Professor Planes,

Thank you very much for submitting your manuscript "BOSO: a novel feature selection algorithm for linear regression with high-dimensional data" for consideration at PLOS Computational Biology.

As with all papers reviewed by the journal, your manuscript was reviewed by members of the editorial board and by several independent reviewers. In light of the reviews (below this email), we would like to invite the resubmission of a significantly-revised version that takes into account the reviewers' comments.

We cannot make any decision about publication until we have seen the revised manuscript and your response to the reviewers' comments. Your revised manuscript is also likely to be sent to reviewers for further evaluation.

Sincerely,

Sergei L. Kosakovsky Pond, PhD

Associate Editor

PLOS Computational Biology

Douglas Lauffenburger

Deputy Editor

PLOS Computational Biology

Feilim Mac Gabhann

Editor-in-Chief

PLOS Computational Biology

Reviewer's Responses to Questions

**Comments to the Authors:**

Reviewer #1: In this manuscript, Valcarcel et al. present a method called Bilevel Optimization Selector Operater (BOSO) to extract meaningful features in linear regression models. With an increase in high-dimensional data in Biology, it has become increasingly challenging to distill meaningful information to the bare minimum, that is sufficient, both for descriptive and predictive models. The identification of the most important variables is done via feature selection, one of the most commonly used tools in machine learning. Valcarcel et al. tackle this issue for linear regression models, and present BOSO as an algorithm that outperforms the other competing approaches in reducing dimensions, and selecting important features. The algorithm utilizes bilevel optimization model, where the split training and validation datasets are used for selection of optimal parameter set (minimize the loss function), and for selection of features with lowest validation error respectively. They perform multiple iterations of feature set, K, and select the best set that results in the optimum model selection criteria. They show that BOSO consistently compares to and/or outperforms the other algorithms in many aspects. They proceed to apply BOSO in predicting drug sensitivity in cancer datasets using RNASeq data. Overall, the prediction from BOSO is improved compared to other methods, while consistently selecting less number of features from the original dataset. Taken together, BOSO appears to generate accurate, and parsimonious models compared to other models for linear regression.

With an increase in multimodal measurements in biology, feature selection has become a critical step in understanding the complex biology, and using information from high-dimensional datasets in palatable way. Overall, I find the manuscript well-written, and easy to follow. However, several issues need to be addressed. My major concern is that I find that the conclusions regarding the superiority of BOSO compared to other methods misleading. BOSO, at most, is comparable to other methods, depending on the parameters used, and only in some instances superior, but very similar to other methods. Some major comments are listed below:

1. Authors stress that the use of validation datasets is novel. Can they show how not having a separate validation dataset undermines the predictive value of feature selection?

2. Authors argue BOSO reduces the computation time. Can they show the computation time differences for different methods used, and how BOSO compares to other methods?

3. For comparison to other methods in Figs 2-6, for many metrics used, BOSO, at most, is comparable to, and not superior than other methods. Authors need to tone down their conclusion to agree with the results.

4. For the use of BOSO in RNASeq data, the discrepancy in model-calculated IC50 and experimental data is huge. It seems to me that BOSO consistently deflates the difference between sensitive, and insensitive cell-lines (Fig. 7d). Compared to experimental data, the prediction suggests that sensitivity is much lower for insensitive cells, while higher for sensitive cells.

5. Based on the results, it seems that some methods are better at predicting sensitivity for insensitive cells, while others are better in predicting sensitivity for sensitive cells. Can authors comment on why that is, and how that information could be used to optimize a better predictive model?

6. Authors state that there is "a significant agreement between computational predictions, and in-vitro experiments." The way the data is presented, it does not seem to have a trend that is consistent for both sensitive and insensitive cell-lines. Can authors elaborate on how they conclude that this is a significant agreement? Some sort of statistical tests might be necessary here.

7. While I agree that feature selection is necessary, reducing the number of features to a very small number might make the model less predictive. To that end, how minimum should the features be? By reducing the features, are we getting rid of valuable information that would otherwise be important for biological processes?

Some minor comments that I have are outlined below:

1. Introduction needs a little more clarity. For instance, "Dimensionality reduction and feature selection selection are the most commonly used strategies to address this issue". What issue? And the transition to linear regression models in the next need to be smoother, as it is, it comes out of nowhere.

2. For gene expression data, can authors elaborate more on how they filtered out the genes with low mean and variance expression? IN methods, the description is not detailed.

Reviewer #2: The proposed BOSO (Bilevel Optimization Selector Operator) is a novel feature selection approach for linear regression. The method is novel and the data analysis using this method shows its advantage compared with other methods, including relaxed LASSO.

The paper is well written and organized, the method is sound, the data analysis is solid. In the Method description, I would like the authors to discuss how to find the optimal values for the parameters M, delta, etc, since these parameters will influence the feature selection results. In all, this is a good work.

Reviewer #3: Valcarcel et al. present BOSO (Bilevel Optimization Selector Operator), a novel method to conduct feature selection in linear regression models. The authors have demonstrated the performance of their method with extensive testing on synthetic data as well as modeling the methotrexate sensitivity of cancer cell lines. The authors have also performed experimental validation of their methotrexate sensitivity predictions for 4 cell lines not included in their training data set. The value of this new approach is two-fold: improved performance (in some but not all synthetic data sets) and increased parsimony (fewer features). The manuscript is mostly focused on analysis of synthetic data, and as such the authors have shown methodological novelty. However, it would be nice to see the authors demonstrate that BOSO can provide new biological insights. Along those lines, there are several issues which warrant further explanation from the authors:

- Page 12: The authors claim that “BOSO obtained the best solution in test data with BIC: mean correlation of 0.628; 2) the models derived from Lasso and Relaxed Lasso have similar mean correlation in test data: 0.636 and 0.631, respectively;”. How is the BOSO solution “best” if the mean correlation of Lasso and Relaxed Lasso is larger than for BOSO?

- Page 13: In the predictive model of MTX IC50 value for “603 cell lines not included in the GDSC database”, what data was included in the training of this BOSO model? Were all 646 cancer cell lines with MTX IC50 values (GDSC) and RNA-seq data (CCLE) used? Or were the data randomly grouped into training (40%), validation (40%) and test (20%) sets like in Figs. 7a and 7b?

- Page 13: In the predictive model of MTX IC50 values for “603 cell lines not included in the GDSC database”, what features were selected by BOSO? The authors claim that BOSO will simplify the interpretation of predictive models by selecting fewer features (see Fig. 7b). Can the authors provide some interpretation of their MTX model? Can the selected features be placed in context of known literature (i.e., the known target of MTX is dihydrofolate reductase), or have the authors uncovered novel predictors of methotrexate sensitivity?

- In “Supplementary Figure 42: Prediction of IC50 values for 50 drugs present in the GDSC database”, what drugs are being tested? I could not find this information, though perhaps it is buried in the Supplemental Data.

- In “Supplementary Figure 42: Prediction of IC50 values for 50 drugs present in the GDSC database”, the mean Pearson correlation coefficient is roughly equivalent for the five methods tested (0.44 – 0.49). I’m worried that averaging the correlation coefficients across all 50 drugs could be confounding the results. If the authors consider each of the 50 drugs individually, how often does BOSO outperform the other numbers? And if the results are mixed (sometimes BOSO is better, sometimes worse), is there any rhyme or reason to when BOSO works best?

Minor points:

- there is a random “S” at the end of the author summary

- page 7, “presentedin”

**Have the authors made all data and (if applicable) computational code underlying the findings in their manuscript fully available?**

Reviewer #1: Yes

Reviewer #2: Yes

Reviewer #3: Yes

PLOS authors have the option to publish the peer review history of their article (what does this mean?). If published, this will include your full peer review and any attached files.

Reviewer #1: No

Reviewer #2: No

Reviewer #3: No
---

## [Decision Letter · Decision Letter 1]

7 May 2022

Dear Professor Planes,

We are pleased to inform you that your manuscript 'BOSO: a novel feature selection algorithm for linear regression with high-dimensional data' has been provisionally accepted for publication in PLOS Computational Biology.

Best regards,

Sergei L. Kosakovsky Pond, PhD

Associate Editor

PLOS Computational Biology

Douglas Lauffenburger

Deputy Editor

PLOS Computational Biology

Reviewer's Responses to Questions

**Comments to the Authors:**

Reviewer #1: Authors have addressed my comments. Good job on the manuscript.

Reviewer #2: The authors have made necessary modifications following the reviewers' comments.

Reviewer #3: The authors have satisfied all my concerns in this revised manuscript.

**Have the authors made all data and (if applicable) computational code underlying the findings in their manuscript fully available?**

Reviewer #1: Yes

Reviewer #2: Yes

Reviewer #3: Yes

PLOS authors have the option to publish the peer review history of their article (what does this mean?). If published, this will include your full peer review and any attached files.

Reviewer #1: No

Reviewer #2: No

Reviewer #3: No

---

## [Editor Report · Acceptance letter]

25 May 2022

PCOMPBIOL-D-21-01162R1 

BOSO: a novel feature selection algorithm for linear regression with high-dimensional data

Dear Dr Planes,

I am pleased to inform you that your manuscript has been formally accepted for publication in PLOS Computational Biology. Your manuscript is now with our production department and you will be notified of the publication date in due course.

With kind regards,

Anita Estes
